# Exploratory Review of the Takotsubo Syndrome and the Possible Role of the Psychosocial Stress Response and Inflammaging

**DOI:** 10.3390/biom14020167

**Published:** 2024-01-31

**Authors:** Niklas Frank, Martin J. Herrmann, Martin Lauer, Carola Y. Förster

**Affiliations:** 1Department of Anaesthesiology, Intensive Care, Emergency and Pain Medicine, Würzburg University, 97080 Würzburg, Germany; 2Center of Mental Health, Department of Psychiatry and Psychotherapy, University Hospital Würzburg, 97080 Würzburg, Germany; herrmann_m@ukw.de (M.J.H.); lauer_m@ukw.de (M.L.)

**Keywords:** Takotsubo syndrome, inflammaging, hypothalamic-pituitary-adrenal axis, sympathetic nerve action, central autonomic nervous system, glucocorticoids, psychosocial stress, psyshological stress

## Abstract

Takotsubo syndrome (TTS) is a cardiomyopathy that clinically presents as a transient and reversible left ventricular wall motion abnormality (LVWMA). Recovery can occur spontaneously within hours or weeks. Studies have shown that it mainly affects older people. In particular, there is a higher prevalence in postmenopausal women. Physical and emotional stress factors are widely discussed and generally recognized triggers. In addition, the hypothalamic-pituitary-adrenal (HPA) axis and the associated glucocorticoid-dependent negative feedback play an important role in the resulting immune response. This review aims to highlight the unstudied aspects of the trigger factors of TTS. The focus is on emotional stress/chronic unpredictable mild stress (CUMS), which is influenced by estrogen concentration and noradrenaline, for example, and can lead to changes in the behavioral, hormonal, and autonomic systems. Age- and gender-specific aspects, as well as psychological effects, must also be considered. We hypothesize that this leads to a stronger corticosteroid response and altered feedback of the HPA axis. This may trigger proinflammatory markers and thus immunosuppression, inflammaging, and sympathetic overactivation, which contributes significantly to the development of TTS. The aim is to highlight the importance of CUMS and psychological triggers as risk factors and to make an exploratory proposal based on the new knowledge. Based on the imbalance between the sympathetic and parasympathetic nervous systems, transcutaneous vagus nerve stimulation (tVNS) is presented as a possible new therapeutic approach.

## 1. Introduction

Takotsubo syndrome (TTS) is a disease associated primarily with emotional and physical trigger factors [1]. Since it is multifactorial, it is better summarized under the term “syndrome”. In the past, TTS has been described in the literature under diverse names, such as “stress cardiomyopathy”, “broken heart syndrome”, or “apical ballooning syndrome” [2]. However, none of these terms accurately describes the ventricular akinesis that may occur in association with this syndrome. Unlike other cardiomyopathies, such as acute coronary syndrome (ACS), which are usually non-transient, TTS is characterized by a transient and reversible left ventricular wall motion abnormality (LVWMA) that can recover spontaneously within hours to weeks.

The most common clinical symptoms are dyspnea, acute chest pain, and/or syncope. However, it is difficult to diagnose TTS based on these symptoms alone. For example, patients with TTS affected by emotional stressors are more likely to present with chest pain and palpitations [3]. Some patients, by contrast, have a clinical manifestation of TTS triggered by severe physical triggers, which is dominated by the manifestation of an underlying acute condition, such as a seizure-triggered or ischemic stroke. In particular, these patients experience less chest pain, which could be associated with neurological complications and impaired consciousness [4,5].

Unfortunately, TTS is very similar to ACS in terms of clinical presentation and ST-segment elevations, making it extremely difficult to distinguish during onset. Approximately 2–3% of patients with the clinical presentation of ACS suffer from TTS [6,7]. Currently, there are no reliable electrocardiogram (ECG) criteria to clearly distinguish ST-segment elevation myocardial infarction (STEMI) from TTS. This makes it even more important to clarify the differential diagnosis of TTS disease. In this case, echocardiography, as an important component of cardiac imaging studies, plays an essential role in the diagnostics of the heart. Cardiac imaging studies should not only be performed as soon as possible during patient presentation but should also be repeated consistently thereafter. Furthermore, cardiac magnetic resonance (CMR) imaging is efficacious in delivering information to differentiate TTS from diseases that cause irreversible myocardial damage, such as myocarditis or myocardial infarction [8].

To date, four major types of TTS variants can be differentiated based on the regional wall motion abnormalities: The most recognized forms are (I) apical ballooning type [9,10], (II) midventricular type, and (III) basal and (IV) focal wall motion patterns [10].

Recent studies focused on the issue of diagnosing TTS by investigating sensitive and specific biomarkers for cardiovascular diseases in blood samples. *MicroRNAs* (*miRNAs/miRs*) have been identified as very promising biomarkers to differentiate STEMI patients from TTS patients [11,12]. In addition to physical and emotional triggers, female sex and age-related deficiency of female sex hormones are known to trigger TTS, and now, inflammation is also being discussed as an emerging pathophysiologic mechanism [1,13].

In this review, we will discuss alternative, novel, unrecognized, and underappreciated potential trigger factors and address the most common diseases associated with TTS. Emphasis will be placed on gender differences, particularly sex hormone differences and aging, and more specifically, on inflammaging as an underappreciated risk factor. In addition, we would like to highlight the possibility of transcutaneous vagus nerve stimulation (tVNS) treatment, as we have shown that stimulation of the parasympathetic nervous system via the vagus nerve (VN) has significant anti-inflammatory effects [14]. The attenuation of low-grade inflammation could be useful in suppressing stress responses and trigger events that could lead to TTS [13,15].

## 2. Brain-Heart Interaction/Axis

The cardiovascular system is associated with regulation throughout the cortical modulation. Although there is still a lack of knowledge on the mechanism of the “brain-heart axis”, various cardiac and neurological diseases have been discussed to be influenced by each other.

The cortical modulation is a network mainly composed of the insular cortex (Ic), anterior cingulate gyrus, and amygdala. This network plays a crucial role in the regulation of the central autonomic nervous system (CAN), e.g., through physical triggers or emotional stressors such as anxiety, excitement, and sadness. Functionally, the Ic can be divided into the right insula, which is associated with sympathetic dominance, meaning that stimulation leads to an increase in heart rate and pressor responses, and the left insula, which is characterized by the parasympathetic tone, leading to a decrease in heart rate and an increase in depressor responses [16]. In addition, the Ic is associated with autonomic, sensory, and motor functions, as well as its bidirectional connections to other brain areas such as the limbic system. Several studies have attempted to demonstrate a link between activation of the Ic and the processing of emotions such as anxiety, fear, anger, panic, and joy [17,18,19,20]. The anterior part of Ic has been shown to be decisively involved in the processing of emotions [21]. This suggests that the processing of emotions by the anterior part of Ic has an influence on the autonomic nervous system and may shift the sympatho-vagal balance toward a sympatico-dominant status. With regard to the current knowledge of the cortical modulation network, with the involvement of the Ic and main parts of the Ic, sympatho-vagal balance is key for homeostasis. A destructed Ic leads to an imbalance, with effects on the cardiovascular system. Its elimination supports the development of TTS. Different factors causing a disruption of the Ic may be a hemorrhage stroke of the middle central cerebral artery, sexual hormones, such as estrogen, or the processing of emotions.

It is known that psychological and physical stressors recruit different brain nuclei to respond to stress. In fact, the activation of the autonomic nervous system (ANS) and neuroendocrine system is central and causes behavioral changes [22]. Studies using functional magnetic resonance imaging (fMRI) of the brain to monitor resting-state functional connectivity demonstrated hypoconnectivity of parasympathetic and sympathetic-associated subnetworks of central brain regions and limbic regions in TTS patients compared to control groups [23].

When self-regulation is disrupted, stress becomes harmful and the body’s susceptibility to diseases such as cardiovascular, psychiatric, and immune disorders increases [24,25]. In the stress response, the locus coeruleus (LC), amygdala nuclei, septal-hippocampal complex, paraventricular nucleus of the hypothalamus (PVH), prefrontal and cingulate cortexes, and parabrachial and raphe nuclei play an important role in the stress response. The resulting signals stimulate the hypothalamic-pituitary-adrenal (HPA) axis. In the signaling chain, the PVH is responsible for the release of the corticotropin-releasing factor (CRF), which induces the release of the adrenocorticotropin hormone (ACTH) by the anterior pituitary gland. ACTH in turn initiates the secretion of glucocorticoids (GC) by the adrenal glands [26,27]. Since CRF can be produced in the central nervous system (CNS), but also in the periphery, it is crucial for the coordination of some physiological systems [28]. Indeed, CRF modulates the stress-induced sympathetic response and is important for the central and peripheral release of norepinephrine (NE) during stress events [29]. This is important to link physiological and behavioral changes and the perception of stress.

Interestingly, in chronic stress situations, a sustained increase in the excitability of the adrenal-medullary axis and the HPA is thought to result in increased NE synthesis [28]. The question arises as to whether the loss of GC self-regulation in the PVH and pituitary gland is of crucial importance in impaired stress management. The loss of GC self-regulation can be explained by the interruption of negative GC feedback and the associated persistent activation and maintenance of elevated systemic GC levels. Due to the higher availability of these hormones, brain structures such as the amygdala and LC enhance the activation of the HPA axis and promote changes in behavior and normal physiology [30,31,32]. Persistently elevated systemic GC levels can trigger immunosuppression and promote the development of autoimmune diseases and mood disorders as well [33].

In this context, chronic unpredictable mild stress (CUMS) should be mentioned. This is an established model that describes mood disorders and stress-induced plasticity of the brain, which are psychological and physical stressors and are caused by a lack of adaptability to various stressful stimuli that are similar to everyday life stressors [34,35]. In this regard, the role of NE release and loss of HPA self-regulation is under discussion [36]. Interestingly, a recent report [13] was able to highlight that TTS patients in different disease phases exhibited the presence of many validated biological and psychological markers of chronic stress as defined in the Trier Social Stress Test (TSST) [37] using blood biosamples from TTS patients: the authors consistently showed the presence of elevated IL-6, TNF-α, NFkB, blood cortisol, DHEA, aldosterone, adrenaline, noradrenaline and dopamine levels; therefore, chronic psychosocial stress as an underlying factor fueling TTS development needs to be acknowledged [38].

## 3. Pathophysiology (Physical and Emotional Triggers)

It is currently believed that TTS is primarily caused by physical and emotional triggers, but psychological and psychosocial stress factors may play a greater role than previously thought. The most important risk factors currently associated with TTS are discussed below.

## 4. Triggers

There is widespread agreement that a major feature of the development of TTS is associated with a stressful event. The most common reason preceding such an event is an emotional or physical trigger. According to some research, physical triggers are more common than emotional stressors, which may also have gender-specific aspects. For example, men seem to be more likely to respond to physical events, whereas women are more likely to be affected by emotional events [9]. Psychological and psychosocial stressors must be identified in this context. Currently, very little is known about the living conditions of patients with TTS. Wallström et al. [38] studied postmenopausal women who were burdened by psychological and psychosocial stress. The patients reported that they felt burdened by responsibility, injustice, and uncertainty long before the onset of Takotsubo syndrome. This long-term stress wore down the respondents’ defenses to such an extent that even the smallest stressors threw them off balance. The results indicate that the social structure of gender can also contribute to the respondents’ condition. These factors may be reflected in the high number of female respondents. By separating the number of cases of TTS patients by gender, the significant difference in the prevalence of TTS between women and men may also be due to the social position and role of women in some countries and cultures. These triggers do not necessarily occur individually, but can also occur as a combination of triggers (e.g., a panic attack or an emotional event following surgery or an accident).

## 5. Emotional Stressors

Emotion is a broad term that refers not only to traumatic emotions, i.e., feelings that arise from traumatic events, interpersonal conflicts, anxiety, fear and anger, earthquakes, or floods [1,39,40,41], but also encompasses positive emotions. Examples include weddings, surprises, and job offers [42]. All of these diverse emotional stressors may be considered triggers for TTS.

As far as stressors are concerned, it is probably not the type of emotion that is decisive, but the harshness of a single event or the combination of several emotions that are insignificant in themselves.

## 6. Physical Stressors

In addition to emotional stressors, physical stressors play an equally important role in the development of TTS. The term physical stressors includes almost any exogenous stress-inducing event. Thus, it includes extremely strenuous activities, medical illnesses (e.g., surgeries [43], traumatic injuries, radiotherapy [44], sepsis [45], or pregnancy [46], to name a few), substance abuse, and nervous system disorders. More specifically, conditions such as head trauma [47], stroke [43], seizures [5], and intracerebral hemorrhage [48] are mainly associated with the onset of TTS.

## 7. Gender Differences in TTS

According to various reports, the severity of TTS is often higher in men than in women. This is in contrast to the prevalence of the disease. Here, women, especially post-menopausal women, are much more frequently affected by TTS [49].

Considering the proportion of gender differences in the USA, Europe, and Japan, although there are varying reports on the proportion of males with TTS, females are predominantly affected by TTS. As TTS is a relatively rare disease, data are currently only being obtained from the US National Inpatient Sample registry [50], The International Takotsubo Registry [9], the Tokyo Cardiovascular Care Unit [49], and the Cardiovascular Research Consortium-8 Universities: CIRC-8U [51]. In fact, the Tokyo Cardiovascular Care Unit claims that prior physical stress is more common in male (50%) than in female (31.3%) patients. In contrast, female patients are more susceptible to emotional stress (male: 19.0% vs. female: 31.0% [49]). According to reports from the International Takotsubo Registry [9], 29.2% of females and about 14.5% of males developed the disease due to emotional stress, whereas 34.3% of females and 50.8% of males developed TTS through physical stress. This is very similar to the Japanese reports [51]. From these reports, it can be assumed that men respond primarily to physical stressors, whereas women tend to respond more to emotional stressors. This is visualized in Table 1, which summarizes the data from the abovementioned reports and compares them on a gender-specific basis.

Although the pathophysiology of TTS is poorly understood, TTS is primarily explained by stress responses, as previously pointed out. This may be due to differences in stress response between the sexes. Interestingly, most patients are postmenopausal women. With regard to their propensity to emotional stress as a trigger for TTS, the effects of estrogen concentration may have a greater impact than previously thought, which requires further investigation.

It is assumed that women show stronger immune responses against foreign and self-antigens. Furthermore, women show a higher prevalence of autoimmune diseases than men. An important role in the activity of immune cells is due to the different attribution to sex hormones between men and women [6,52]. In experimental rat models, the Cidlowski group was able to show that males and females show a difference in the prevalence of many major diseases that are attributable to inflammatory components. Interestingly, a link between inflammatory diseases and the sexually dimorphic effects of glucocorticoids may be important for the sex-specific differences in prevalence. Based on the outcome of these studies, the anti-inflammatory effects of *glucocorticoid receptors (GR)* appear to be more effective in men, whereas a lack of *GR* may promote certain diseases in women. This has been documented in vivo in the liver in a sepsis model of systemic inflammation [53]. This indicates that a primary mechanism in homeostatic female mice ensures a faster response to inflammatory stimuli and thus causes a stronger expression of the most common proinflammatory genes.

In addition to emotional stress, behavioral stress reactions, psychological stress, and estrogen concentration can also be an important trigger in postmenopausal women. Studies have shown that there are significant differences between pre- and postmenopausal women in their reactions to psychological stress. Importantly, estrogen appears to attenuate the effect of stress-induced reactions. This means that in TTS, the stress response could be exacerbated as there is an imbalance in androgen/estrogen levels [54].

In addition, the dexamethasone/corticotropin-releasing hormone (Dex-CRH) test indicates, that the negative feedback of the HPA axis is altered in older women. This is shown by studies of psychological and endocrine responses to psychosocial stress and Dex-CRH in healthy postmenopausal women and young controls. In addition, the current data suggest that estradiol supplementation appears to modulate HPA feedback sensitivity in humans [55].

## 8. How Does Estrogen Concentration Affect Postmenopausal Women?

Considering the much-discussed gender differences in TTS, the number of postmenopausal women with TTS is alarming. This raises the question of the importance of sex hormones and their influence on the development of TTS.

In this context, a major focus lies in the possible role of estrogen deficiency in TTS. The severe effects of a reduction in estrogen levels are underlined by the high number of postmenopausal women who develop TTS. An animal study investigated two groups of female rats: group 1 comprised ovariectomized (OVX) female rats, while group 2 consisted of ovariectomized rats supplemented with estradiol (OVX + E). They were all subjected to immobilization stress to evaluate cardiac changes. In the OVX rats, contraction of left ventriculography was significantly reduced, whereas no sizeable responses were observed in the OVX + E rats. In addition, both groups exhibited a significantly increased heart rate, although the heart rate of the OVX rats was higher [56,57]. Rivera et al. [58] showed that patients who underwent bilateral oophorectomy, as opposed to unilateral oophorectomy, had a higher mortality rate due to cardiovascular events. By analyzing both results, it can be concluded that estrogen substitution may positively influence the risk of developing cardiovascular events due to emotional stress. Currently, estrogen is thought to have a cardioprotective effect and suppress the sympathetic nervous system. Evidence for the cardioprotective effect of estrogen can be found, for example, in a study by Brenner et. al. In the study, postmenopausal women with TTS were compared with age- and sex-matched patients with myocardial infarction (MI) and patients with normal coronary arteries. The aim was to investigate the different influences of sex hormones (estradiol (E2), progesterone (P), luteinizing hormone (LH), and follicle-stimulating hormone (FSH)) during onset and long-term follow-up. Interestingly, E2 levels were significantly higher in TTS patients at hospital admission, whereas no changes were observed in MI patients and the control group. This suggests that elevated E2 concentrations have a cardioprotective effect [59].

There are different mechanisms by which estrogen concentration in postmenopausal women may influence the development of TTS. Postmenopausal women display a significant reduction in estrogen concentration which is thought to accelerate aging and increase the risk of mortality from cardiovascular events [58]. This information is supported by investigations of the Multi-Ethnic Study of Atherosclerosis (MESA), which reported a 12-year follow-up of 2834 menopausal women with an increased testosterone/estradiol ratio, highlighting the incidence of heart failure with reduced ejection fraction (HFrEF) [60]. Currently, the role of estradiol in the regulation of energy metabolism is being discussed [61]. An animal study by Zhu et al. [62] revealed new insights into the regulation of blood pressure and vascular function, in which *estrogen receptor alpha (ERα)* and *ERβ* play a central role. *ERs* were confirmed to be expressed in blood vessels, where they influence vasoconstriction. In addition, it has been shown that persistent systolic/diastolic hypertension developed in aging *ERβ-deficient mice*. Furthermore, estrogens have been observed to reduce the sympathetic response to psychological stress and reduce catecholamine-induced vasoconstriction [63,64]. Lastly, endothelial nitric oxide (NO) synthase can be influenced to modulate vasomotor tone [65]. Whether estrogen has exclusive indirect effects (via the nervous and/or vascular systems) or also has direct effects on the cardiomyocyte has not yet been conclusively determined. In the study by Förster et al. [66], *ERβ (−/−) mice* display myocardial disarray, disrupted intercalated discs, profound changes in nuclear structure, and increased number and size of gap junctions. However, no ERβ was found in the myocardium, which strengthens the hypothesis that estrogen acts indirectly on the myocardium.

Although there are differences in the clinical presentation of HFrEF and TTS, low estrogen levels could be considered a common risk. This could be supported by the fact that postmenopausal status is an underlying pathophysiological feature in both TTS and HFrEF. To better understand the pathological mechanism and the influence of sex hormones, further studies need to be performed with both male and female case groups.

Another interesting aspect to investigate is the functional cerebral asymmetry, in particular, the influence on Ic activity in the modulation network. As discussed earlier, we can distinguish between a right cortex associated with sympathetic activity and control of emotional stimuli and the left hemisphere dominated by the parasympathetic tone of the heart [42,67]. Estrogen in particular has been found to cause a correlation with left hemisphere activation [68]. This leads to the suggestion that low estrogen levels promote left hemisphere inactivation. It can be concluded that low estrogen may shift the sympathovagal balance toward sympathetic activity. This can be explained by the different associations of the hemispheres to the sympathetic and parasympathetic nervous systems.

As previously discussed, low estradiol levels in postmenopausal women have been named as a possible promoter of TTS. Studies have shown that low E2 is associated with a higher risk of developing TTS. In this context, the responsiveness of postmenopausal women to the effects of age and estrogen on stress needs to be closely examined.

As noted earlier, HPA function may play a greater role in chronic stress as well as in the behavioral response to stress. In this context, one study has examined HPA function as influenced by circulating estradiol levels and hormonal status during the stress response. In relation to psychosocial stress, the results showed an enhanced response of the HPA axis during the low-estrogen phase of the menstrual cycle compared to the high-estrogen phase and additionally during menopause [54]. In their studies, Lindheim et al. [54] were able to demonstrate significant differences in the reactions to psychological stress between premenopausal women and postmenopausal women. They pointed out that this could also be a reason for the higher prevalence of cardiovascular disease in women. This is a very interesting but also alarming indication that postmenopausal women may be more affected by psychosocial stress than men due to low estrogen levels.

## 9. Sympathetic Nervous System

The pathophysiological mechanisms of TTS have not been fully elucidated to date. Studies strongly suggest that it is closely related to sympathetic nervous system stimulation. A significant proportion of patients who experience TTS can be attributed to emotional and physical stressors [69,70,71]. The stress response is mediated by anatomical structures of the central nervous system and various peripheral organs. In this context, emotional stressors, for example, can lead to brain activation followed by an increase in concentrations of cortisol, epinephrine, and NE.

The neocortical complex and the limbic system, which is responsible for classifying events as “threatening,” lead TTS patients to overreact to physical or emotional triggers. This is followed by the stimulation of the sympathetic nervous system (SNS). Sympathetic activation can be explained by two neurohumoral axes: (1) The sympathetic-adrenal axis is activated by immediate stressors and is characterized by catecholamine release from the adrenal medulla. It must be noted that adrenal medullary catecholamine release is an essential component of the neuroendocrine stress response axis. (2) In contrast, the HPA is increasingly activated by chronic stressors. This is caused by a continuous release of cortisol from the adrenal cortex [72].

The stress response is a complex process involving multiple pathways and components, including behavioral, endocrine, and autonomic changes that lead to a coordinated response [22]. Some of these pathways are associated with the activation of the hypothalamic-pituitary-adrenal axis [26,27]. In individuals with inadequate self-regulation, stress can have detrimental effects and cause pathologies [25], e.g., through excessive inflammation. Interestingly, the animal study presented here, in which male rats were exposed to CUMS, showed increased expression of *GR* in response to CUMS. In this study, Malta et al. [36] highlighted that *GR* plays a role in fine-tuning CUMS responses, which have been shown to depend on GC and NE signaling in male rats. Specifically, 14 days of CUMS administration was shown to induce sustained hyperactivity of the HPA axis in male rats. This was reflected in an increase in plasmatic corticosterone and adrenal hypertrophy, both of which were dependent on increased GC and NE release triggered by each stress session. CUMS exposure also increased *CRF2 mRNA* expression and *GR* protein levels in basic brain structures related to HPA regulation and behavior [36]. This rationale was reinforced by the observation that repeated stress (CUMS) correlates with increased *GR* expression in the spleen [73]. GC are stress-induced steroids that not only have inflammatory and immunosuppressive effects but also regulate the function and development of the central nervous system (upregulation of sympathetic nerve activity and downregulation of the parasympathetic drive), intermediary metabolism, vascular tone, and above all, the process of programmed cell death [74,75]. A consistent trend toward increased GR levels has been observed in the serum of TTS patients [13].

The focus is on the fact that the activation of the HPA axis and the SNS, which play an important role in TTS, is associated with increased peripheral proinflammatory markers in the blood, especially in chronic stress [76]. Currently, there is little empirical evidence that peripheral levels of glucocorticoids and/or catecholamines mediate this effect. An established model is of significance here, in which chronic stress leads to GR resistance, which in turn results in an upregulation of GR levels and a lack of downregulation of the inflammatory response.

Currently, it is thought that cellular sensitivity to these ligands may contribute to the inflammatory mediators that accompany chronic stress. A link between chronic stress and the sensitivity of the GR and the *β-adrenergic receptor (β-AR)* has been hypothesized. Studies have shown that glucocorticoid resistance and *β2-AR* signaling pathways promote peripheral proinflammatory states associated with chronic psychological stress [77]. Interestingly, several research studies present similar results with social stressors in mice, primates, and humans as chronic stress is associated with upregulation of pro-inflammatory gene transcription. Notably, a significant downregulation of *GR* sensitivity was observed, which could lead to increased GR expression. These stress-related findings are also associated with an atypical intracellular *β-AR* signaling pathway. However, its significance in TTS still requires further investigation [77,78,79,80,81,82]. *GR* expression has been described as a suitable surrogate marker in various stressful situations, such as post-traumatic stress disorder or the observed glucocorticoid resistance in chronic stress [83].

As explained earlier, in most cases TTS is triggered by an emotional or physical event, but diseases such as pheochromocytoma or other central nervous system disorders closely associated with catecholamine excess can also cause TTS-like dysfunction and should be considered. More detailed investigations as they relate to TTS are essential. Intravenous administration of catecholamines and beta-agonists to induce TTS and other ballooning patterns is clear evidence. Several animal studies suggest that adrenergic activation plays an important role in the development of TTS [84,85,86]. The increased level of norepinephrine in the coronary sinus of TTS patients due to an increased local release of myocardial catecholamines supports the important influence of sympathetic stimulation [87]. Another aspect supporting the influence of sympathetic stimulation is the analysis of heart rate variability. During the acute phase, a suppression of the parasympathetic activity is observed [88]. This is in line with some microneurographic studies, demonstrating a decreased baroreflex control in TTS patients [89].

Although there is now strong evidence that sympathetic activation plays a central role in TTS, the effects of catecholamine excess on the various ballooning patterns characteristic of TTS have not been truly elucidated. So far, several reasonable suggestions have been proposed. For example, it has been suggested that a catecholamine surge may lead to myocardial damage. In this context, several mechanisms can be mentioned, such as adrenoreceptor-mediated damage and microvascular coronary vasoconstriction, which increase cardiac work [90].

## 10. Inflammaging

Inflammaging is a term that has become associated with aging and age-related diseases. It is a low-grade, sterile, and chronic inflammation that worsens with age. This contributes to the pathogenesis of age-related diseases. Inflammation is an important characteristic of aging and other comorbidities associated with age-related decline, such as neurodegeneration, Alzheimer’s disease, age-related macular degeneration, age-related hearing loss, and type 2 diabetes.

Dysregulation of the immune system is strongly associated with aging, and obvious features are high blood levels of pro-inflammatory markers, e.g., interleukin (IL)-1, -6, -8, tumor necrosis factor (TNF), and C-reactive protein (CRP) [91,92,93,94]. One of the most important features associated with aging is a decrease in autophagy mechanisms [95]. Autophagy describes the cellular housekeeping mechanisms to eliminate dysfunctional intracellular proteins that primarily prevent the activation of inflammatory responses. In contrast, the consequence of an age-related decline in autophagy mechanisms is increased activation of inflammatory responses and an increase in pro-inflammatory markers [96].

Another important component that affects inflammation is the *telomeres, repetitive deoxyribonucleic acid (DNA)* sequences located at the end of chromosomes to protect them from fusion or decay. However, they shorten with each cell division and can eventually lead to senescence and death. For example, a study from the Health, Ageing and Body Composition Study [97] showed a correlation between shorter telomere lengths and inflammatory markers such as increased IL-6 and TNF [97,98]. Other studies have shown a correlation with CRP [99] and chronic inflammatory diseases of the lung and kidney [100]. In summary, this means that there is likely a link between chronic stress and aging. This suggests that people of advanced age are generally affected by low-grade chronic inflammation, and are thus at increased risk of triggering TTS by physical or emotional trigger factors, as well as by CUMS. An age-related decrease in the immune response and increased proinflammatory blood levels are likely to be responsible for older people being more susceptible to TTS. Based on current knowledge, “inflammaging” seems to be very well described under the term of age-related low-grade sterile chronic inflammation. From our knowledge of the pathophysiology of TTS, we can see similarities to inflammaging that indicate a link between the two. This is especially the case given the age distribution of TTS patients and the elevated proinflammatory markers in blood samples from hospitalized TTS patients [13]. In particular, with regard to emotional or physical triggers, low-grade chronic inflammation could be crucial for the onset of TTS as it mainly affects people between 60 and 80 years of age. Since TTS seems to be correlated with age-related chronic inflammation, a new treatment option using tVNS can be investigated. As we have shown in our recent research, there is a strong correlation between chronic inflammation and hyperactivity of the SNS. Anti-inflammatory results have been observed when stimulating the parasympathetic nervous system [14], suggesting that the stimulation of the VN may also have beneficial effects on the prevention of TTS, a stress-induced cardiomyopathy.

Although there are still very little data on inflammation associated with TTS, it should not be underestimated as a risk factor and further investigation should be considered.

## 11. Conclusions

The latest evidence suggests that CUMS plays a greater role in the development of TTS than previously thought [73]. In particular, it concerns the link between chronic psychological stress and a decrease in *GR* and *β-AR* sensitivity. Resistance leads to overexpression of GR and dysregulation of the HPA axis. The dysregulated negative feedback leads to increased GC levels and long-term immunosuppression, with increased levels of peripheral pro-inflammatory markers (Figure 1). This leads to increased activity of the sympathetic nervous system and reduced activity of the parasympathetic nervous system. At this point, tVNS treatment can be very effective. Frank et al. [14] showed that tVNS has a positive effect on parasympathetic activity and plays an important role in reducing chronic pro-inflammatory markers. However, as far as its applicability is concerned, clinical studies and results are still lacking and should be the subject of future research.

## Figures and Tables

**Figure 1 biomolecules-14-00167-f001:**
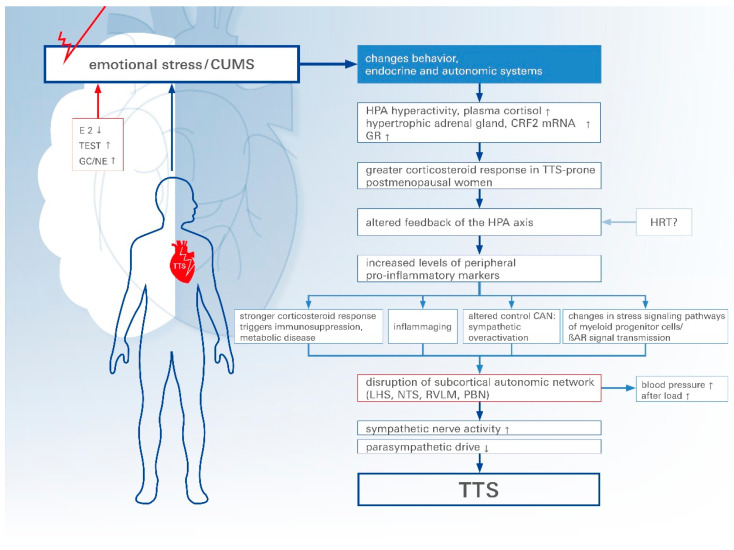
The potential contribution of chronic stress and inflammaging to TTS susceptibility. A novel concept for TTS development, involving underlying chronic stress, glucocorticoid resistance, altered immune response culminating in inflammaging, and altered control CAN and *βAR* signaling. The integration of these effects might enhance susceptibility to TTS following an acute trigger. CUMS: chronic unpredictable mild stress, E2: estradiol, TEST: testosterone, GC/NE: glucocorticoid/norepinephrine, HPA: hypothalamic-pituitary-adrenal, GR: glucocorticoid-receptor, HRT: hormone replacement therapy, CAN: central autonomic nervous system, LHS: limbic-hypothalamic system, NTS: nucleus tractus solitarii, RVLM: rostral ventrolateral medulla, PBN: parabigeminal nucleus, and TTS: takotsubo syndrome. Arrow up = upregulated, Arrow down = downregulated.

**Table 1 biomolecules-14-00167-t001:** Systematic Overview of TTS Trigger Factors on a Gender-Specific Basis.

Country	Registry	Study Period	Age		Preceding Stress						Reference
					Emotional Stress		Physical Stress		Absence of Stress		
			Female	Male	Female [%]	Male [%]	Female [%]	Male [%]	Female [%]	Male [%]	
USA	NI Sample	2009–2010	66.2	59.2	-	-	-	-	-	-	[50]
USA/Europe	Inter TAK Registry	1998–2014	66.8	62.9	29.2	14.5	34.3	50.8	28.8	25.7	[9]
Japan	Tokyo CCU Network	2010–2012	76	72	31.0	19.0	31.3	50.0	37.7	31.0	[49]
Japan	CIRC-8U	1997–2014	71.5	71.8	26.0	10.0	46.0	64.0	28.0	26.0	[51]

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
