# Peer review of "Exploratory Review of the Takotsubo Syndrome and the Possible Role of the Psychosocial Stress Response and Inflammaging"

_biomolecules, 2024, doi:10.3390/biom14020167_

Round 1

Reviewer 1 Report

Comments and Suggestions for Authors

In this review, Frank and co-authors provide a comprehensive overview of Takotsubo syndrome (TTS), a cardiomyopathy caused by physical and/or emotional stresses, with a focus on the possible role of psychosocial stress and inflammaging. In particular, they delve into novel potential triggers that could play a role in the pathophysiology of the disease, exploring their connection to sympathetic hyperactivity observed in many cardiovascular diseases, including TTS. Ultimately, they propose transcutaneous vagus nerve stimulation as a new possible therapeutic approach for patients with TTS. 

The main strength of this review is that it is the first review on Takotsubo syndrome taking into consideration a wide range of aspects contributing to the onset of the disease, from the conventional and well-known physical and emotional triggers to psychological/psychosocial stress conditions, inflammation, aging, and gender and sex hormones differences. It is also the first review that proposes transcutaneous vagus nerve stimulation as a possible therapeutic approach for patients with TTS. The authors made good use of the current literature. For these reasons, I believe that this review will be of interest to many readers.

The main limitation is the lack of clarity in some parts of the text, which makes it sometimes difficult to read and interrupt the flow of the story. Indeed, some sentences are not complete (see detailed report below) and a couple of times the authors refer to “several studies” when they only cite one paper. 

I offer the following suggestions for the authors to consider when revising the text and figure/table (the same detailed reviewer report is attached).

Abstract, Line 12: Physical and emotional stresses are widely discussed…

Abstract, Line 18-19: In the following review, these risk factors are…

Page 1, Line 41: “…whereas patients triggered by physical stressors are more likely to clinically manifest an acute illness.” I’d suggest the authors explicitly mention the type of acute illness manifested by TTS patients and provide references to support their statement.

Page 2, Line 46: “ST-segment elevation myocardial infarction (STEMI)…” I would add here the abbreviation for ST-segment elevation myocardial infarction, STEMI, rather than later at line 62.

Page 2, Line 57: “The most recognized forms are I) apical ballooning type (7, 8), II) midventricular type, III) basal and IV) focal wall motion patterns (8).” There is no need for this sentence to be in a new paragraph since it refers to the previous one. 

Page 2, Line 62: “very promising biomarkers to differentiate STEMI…”. If the authors decided to abbreviate 'ST-elevation myocardial infarction' earlier in line 46, here they can use the abbreviation.

Page 2, Line 74. “Attenuation of low-grade inflammation could be useful in suppressing stress responses and trigger events that could lead to TTS (13-15).”  I'm not sure to understand how reference 15 (Karnati S et al., 2018) is related to how attenuating inflammation can improve TTS outcomes. Reference 15 refers to lipidomic analysis on mouse lungs using mass spec.

Page 2, Line 79-80. “High levels of the named pro-inflammatory markers are seen in the majority of elder people, even without acute diseases.” In my opinion, it is hard to contextualize this statement with the brain-heart axis. I think this sentence can be more appropriate somewhere else.

Page 2, Line 84. “…emotional stressors such as    anxiety, excitement, and sadness.” Remove extra space. 

Page 3, Line 131. “…the adreno-medullary axis…”. Remove the dash, or use the adrenal-medullary axis. 

Page 3, Line 147. “Interestingly, recent reports (13) were able to highlight that TTS patients in different…”. If there is more the one report highlighting that TTS patients display many validated biological and psychological markers of chronic stress, the authors should refer to them.

Page 3, Line 148. “…validated biological and psydhological psychological markers…”

Page 4, Line 180-181. “Emotion is an open term that includes not only traumatic emotions, which are feelings based on traumatic events.” This sentence seems incomplete. It might be a good option to rephrase it by specifying the range of emotions included in the term 'Emotion'.

Page 5, Line 214. “In contrast, female patients are more addicted to emotional stress (male: 19.0% vs. female: 31.0%, (= 0.039)) (50).” I'm not sure "addicted" is appropriate here. What about: "In contrast, female patients are more susceptible to emotional stress"?

Page 5, Table. The authors did not cite the table anywhere in section 7. In addition, the table does not have a number, a title, and a legend. 

Line 219. This sentence can be a good place where to cite the table. “From these reports, it can be concluded that men respond primarily to physical stressors, whereas women tend to respond more to emotional stressors.”

Page 6, Line 230. “Furthermore show women a higher prevalence to autoimmune diseases than compared to men.” This sentence is incomplete and needs a subject/s. 

Page 6, Line 232. “It is assumed that women show stronger immune responses against foreign but also 229 against self-antigens. Furthermore show women a higher prevalence to autoimmune dis-230 eases than compared to men. An important role in the activity of immune cells is due to 231 the different attribution to sex hormones between men and women (4, 53).” I'm not sure how reference 53, a comment to the authors of "Is high dose catecholamine administration in small animals an appropriate model for Takotsubo syndrome?", is closely related to this paragraph.

Page 6, Line 242-244. “This suggests that a priming mechanism in homeostatic female animals ensures a faster response to inflammatory stimuli, so that the expression of the most frequently regulated proinflammatory genes is induced to a greater extent in female mice.” I find this sentence somewhat challenging to comprehend. Could the authors consider rephrasing it for better clarity?

Page 6, Line 264-265. “group 1 were ovariectomized (OVX) female rats and group 2 were those that were ovariectomized but also supplemented with estradiol (OVX+E).” For improved readability, the authors can consider a slight rephrase: "group 1 comprised ovariectomized (OVX) female rats, while group 2 consisted of ovariectomized rats supplemented with estradiol (OVX+E).”

Page 6, Line 279. “…postmenopausal women with TTS were compared with age- and sex-matched patients with myocardial infarction (MI) and patients with normal coronary arteries.”

Page 7, Line 286-287. “There are different approaches to how estrogen concentration in postmenopausal women may influence the development of TTS.” To improve readability, the authors can consider a slight rephrase: "There are different mechanisms by which estrogen concentration in postmenopausal women may influence the development of TTS."

Page 7, Line 288-290. “In postmenopausal women, estrogen concentration is significantly reduced and is supposed to accelerate aging and enhance the rate of mortality due to cardiovascular events (59).” To enhance clarity and flow, the authors can consider a slight rephrase: "Postmenopausal women display a significant reduction in estrogen concentration which is thought to accelerate aging and increase the risk of mortality from cardiovascular events."

Page 7, Line 297. “ERs were confirmed to express blood vessels and influence vasoconstriction.” Do the authors mean that "ERs were confirmed to be expressed in blood vessels where they influence vasoconstriction."? If so, please correct appropriately.

Page 7, Line 306-309. “In the study by Förster et al. (67), myocardial disarray, disrupted intercalated discs, profound changes in nuclear structure, and increased number and size of gap junctions were observed, but no ERb was found in the myocardium (67) which strengthens the hypothesis that estrogen acts indirectly on the myocardium.” In the mentioned study, the aforementioned ultrastructural alterations of the myocardium were observed in ERb -/- mice. The sentence does not specify it. The authors can say "In the study by Forster et al., ERb -/- mice display myocardial disarray, disrupted intercalated discs, ... but no ERb was found in the myocardium, which strengthens the hypothesis that estrogen acts indirectly on the myocardium."

In addition, there is no need to cite two times the same reference within the same sentence.

Page 7, Line 307. “In the study by Förster et al. (67), myocardial disarray, disrupted intercalated discs…”. 

Page 7, Line 328. “In this context, the responsiveness of older women to the effects of age and estrogen on stress needs to be closely examined:” Need to close the period. 

Page 8, Line 383. “The focus is on the fact that activation of the HPA axis and the sympathetic nervous system SNS,…” Since the sympathetic nervous system was already abbreviated earlier, the authors can use the abbreviation- SNS.

Page 9, Line 391. “Currently, attention is focused given on the possibility…”

Page 9, Line 395-397. “Studies have shown that glucocorticoid-resistance associated with and β2-adrenergic receptor signaling pathways promotes peripheral proinflammatory states associated with chronic, psychological stress (78).”

Page 9, Line 398-403. “Interestingly, research shows similar results with social stressors in mice, primates and humans, as chronic stress is associated with upregulation of pro-inflammatory gene transcription. Notably, a significant downregulation of GR sensitivity was observed, which could lead to increased GR expression. These stress-related findings are also associated with an atypical intracellular β-AR signaling pathway. However, its significance in TTS still requires further investigation (78-83).” Since many research studies are cited, the authors should write “Interestingly, several research studies…”.

Page 9, Line 412. “Several animal studies suggest that adrenergic 412 activation plays an important role in the development of TTS (85).” The authors could cite more animal studies as they mentioned in their sentence.

Page 9, Line 422. “So far, several suggestions have been made that seem reasonable.” This sentence can have a better flow. The authors could say: "So far, several reasonable suggestions have been proposed."

Page 10, Line 463. “…there is a strong correlation between chronic inflammation and hyperactivity of the sympathetic nervous system SNS.” 

Page 10, Line 464. “While anti-inflammatory results were observed when the parasympathetic nervous system was stimulated (12).” This sentence can't exist by itself. The authors could fuse this sentence with the sentence thereafter.

Page 10, Line 470. “The latest findings suggest that CUS/CUMS plays…”. The authors never spell out CUMS as chronic unpredictable mild stress in the text. It is also not clear why the authors chose to use CUS in section 2 and CUMS in section 9. Please explain or use only one of the terms if they are interchangeable.

Page 10, Line 483. “This may be a further indication that women have an increased prevalence of cardiovascular diseases.

Page 10, Line 488. “This is exactly the point at which where tVNS treatment comes into question play.” I think "into play" might sound more appropriate as it suggests that tVNS treatment becomes relevant at this point and it should not be in discussion.

Page 10, Line 490. “…plays an important role in the depression reduction of chronic pro-inflammatory markers.”

Figure 1. It might be useful to include in the legend the full spellings of the abbreviations used in the figure for easier comprehension by the readers. 

Comments on the Quality of English Language

There are some parts where it is a little difficult to understand, and further editing for English grammar, word usage, and sentence structure would be helpful. 

The authors can find some recommendations in the detailed reviewer report above.

Reviewer 2 Report

Comments and Suggestions for Authors

Upon careful review, the following critical feedback and improvement suggestions have been identified for the review paper.  These suggestions aim to enhance the overall clarity and coherence of the review.

The abstract provides a good overview of the review topic. However, it would be beneficial to include a concise statement of the review's objectives and anticipated outcomes. Additionally, the abstract should be structured to provide a clear understanding of the scope and relevance of the review to potential readers. Consider explicitly stating the review's objectives and anticipated outcomes in the abstract to provide a clearer understanding of the review's focus and potential contributions. The abstract could benefit from a clear summary of the potential novel trigger factors and diseases associated with TTS to give readers a better understanding of the paper's focus. 

Emotional Stressors: The section on emotional stressors is comprehensive, providing a detailed overview. However, it would benefit from supporting references for the mentioned emotional stressors

Brain-Heart Interaction/Axis: This section provides a detailed exploration of the brain-heart interaction and the role of the central autonomic nervous system. However, the presentation lacks cohesion and clarity. The information seems fragmented and could benefit from a more organized structure

Gender Differences in TTS: The section on gender differences in TTS provides valuable insights. However, the information is presented in a rather cumbersome manner, making it challenging for readers to extract key findings.

Specific improvements are needed to streamline the content and clarify the reported proportions of males and females developing TTS from different sources (Lines 199-228)

The connection between age-related inflammaging, chronic stress, and TTS needs to be spelled out more explicitly, with a focus on the biological and physiological processes that link age-related chronic inflammation to the development of TTS

The excessive number of paragraphs could potentially challenge reader comprehension. 

It is recommended to streamline the content for usability. Furthermore, the USA NI Sample 2009–2010 data in the table should be substituted with more recent studies that address gender disparities. 

As for the conclusion, condensing it to a brief 5-6 lines with no citations or references would enhance its clarity and impact.

Comments on the Quality of English Language

Minor corrections required

Round 2

Reviewer 1 Report

Comments and Suggestions for Authors

Thank you to the authors who addressed well both reviewers’s comments and suggestions. Unfortunately, there are some parts where adjustments are made, and other errors are introduced. Please, carefully read the manuscript to ensure overall clarity and comprehensibility. 

I offer the following suggestions for the authors to consider when revising the text. 

Abstract, Line 15: “This review focuses in particular on postmenopausal women…”

Abstract, Line 26-29: This sentence is too long, making it difficult to follow. Please rephrase for clarity. 

Page 2, Line 58: “This makes resolving the differential diagnosis in TTS disease that much more significant.” Please, correct this. 

Page 4, Line 150: “Interestingly, a recent report (15) were was able to highlight that…”

Page 6, Line 230. This sentence is still without subject/s. Please correct. “Furthermore show women a higher prevalence to autoimmune diseases than compared to men.” 

Page 7, Line 273. The first sentence needs a period. “Currently, estrogen is thought to have a cardioprotective effect and to suppress the sympathetic nervous system: Evidence for the cardioprotective effect of …“

Page 8, Line 302-305. The sentence has been rephrased but there are still errors. I would suggest the authors carefully read when adjustments are made. “In the study by Förster et al. (67), ERβ (-/-) mice) display myocardial disarray, disrupted intercalated discs, profound changes in nuclear structure, and increased number and size of gap junctions were observed, but no ERβ was found in the myocardium, which strengthens the hypothesis that estrogen acts indirectly on the myocardium.”

Page 9, Line 388. Remove comma. “…chronic, psychological stress (78).” 

Comments on the Quality of English Language

See the reviewer's report above.

Reviewer 2 Report

Comments and Suggestions for Authors

The excessive number of paragraphs is still a challenge. The conclusion would benefit from a concise approach. Additionally, refraining from adding new citations in the conclusion would enhance its focus and clarity.

Comments on the Quality of English Language

I have noticed some grammatical errors that still need to be addressed.
